# Gold Recovery from Sulfide Concentrates Produced by Environmental Desulfurization of Mine Tailings

Olivier Allard [1,2], Mathieu Lopez [1,3], Isabelle Demers [1,*] and Lucie Coudert [1]

1   Research Institute on Mines and Environment (RIME), Université du Québec en Abitibi-Témiscamingue (UQAT), Rouyn-Noranda, QC J9X5E4, Canada
2   Department of Chemistry, University of Sherbrooke, Sherbrooke, QC J1K2R1, Canada
3   SLN—Société le Nickel, Nouméa 98848, New Caledonia
*   Correspondence: isabelle.demers@uqat.ca; Tel.: +1-819-762-0971 (ext. 2343)

**Abstract:** Environmental desulfurization is gaining attention as an efficient approach to manage sulfidic mine tailings by separating sulfide minerals from tailings. While extensive research has been done to reuse desulfurized tailings in mine site reclamation, the responsible management or potential of valorization of desulfurization concentrates did not receive as much attention. The objective of this study was to evaluate the potential to recover Au from desulfurization concentrates originating from active gold mine sites. Desulfurization concentrates were produced by froth flotation of tailings and characterized. Cyanidation tests and gravity separation experiments were performed on the concentrates. Desulfurization concentrates, mostly composed of fine particles, contained high amounts of S (between 23.0 and 34.5% S) and variable contents of Au (between 0.7 and 1.9 g/t). Mineralogical characterization by SEM showed that 99% of Au-bearing particles were locked, mainly in pyrite (94%). Cyanidation allowed the recovery of 13 to 78% of Au, depending on the desulfurization concentrate. Low Au-bearing particles liberation, and possibly the presence of Cu, were identified as parameters negatively affecting the Au recovery. Gravity separation was poorly efficient (<50%) in recovering Au, which is quite consistent with physical (small particle size) and mineralogical characterizations. A desulfurization concentrate reprocessing flowsheet is proposed.

**Keywords:** environmental desulfurization; cyanidation; gravity separation; gold recovery; mine tailings





## 1. Introduction

Mine tailings consist of low-value and finely grained waste material that are produced by the ore beneficiation process, where valuable minerals (or metals) are separated and concentrated. They are deposited as a slurry in tailings storage facilities. Sulfidic mine tailings are prone to oxidation by the reaction between sulfide minerals such as pyrite, atmospheric oxygen, and water [1]. Acid mine drainage (AMD), which consist of a low pH, high sulfate and metal(loid) content effluent, results from sulfide oxidation in the absence of sufficient amounts of neutralizing minerals (e.g., carbonate and/or silicate minerals) [2,3]. While AMD treatment processes are available to mitigate its environmental impacts, prevention of oxidation is generally the preferred approach for long term AMD management [1].

Environmental desulfurization has been developed as a method to reduce the AMD potential of mine tailings [4–6]. The objective of environmental desulfurization is to remove enough sulfides from the tailings by separating the tailings into two fractions: (i) desulfurized tailings that are non-acid generating; and (ii) a sulfide rich sulfide concentrate. Flotation provides the mean to perform the separation, however other methods such as gravity separation may be used in some cases [7]. Environmental desulfurization by flotation consists of a process that involves the bulk flotation of sulfides still present in the tailings after mineral processing. The flotation target is therefore related to the sulfide

content of the tailings, and also to their neutralization potential. Tailings with relatively high neutralization potential (provided by carbonate and/or silicate minerals) can have a small amount of sulfides without being acid-generating. Indeed, when these sulfide minerals oxidize, the acidity produced is neutralized by carbonates/silicates. On the other hand, tailings with a low neutralization potential require a drastic reduction in sulfide content to be considered non-acid generating [1].

Environmental desulfurization is usually proposed to be included at the end of the beneficiation process before tailings deposition [8,9]. In the past years, research focused on the optimization of the desulfurization process for different types of sulfide minerals [10–13]. Sulfide flotation is optimized considering several parameters such as the types of sulfides and sulfosalts present, the neutralization potential, the state of mineral surfaces as well as the slurry physico-chemical parameters [14–18]. For example, 95% of pyrrhotite was removed from a nickel mine tailings, producing desulfurized tailings that are non-AMD generating [12]. A porphyry-copper sulfide tailings was desulfurized from 9 to 0.2% S, and was tested as non-acid generating after desulfurization by flotation [15].

Once separated, desulfurized tailings and sulfide concentrate can be managed separately to reduce environmental risks, or ideally should be valorized. Extensive research work was performed, and is still in progress, on the valorization of desulfurized tailings [19–24]. Being low-AMD risk material, desulfurized tailings can be viewed as silty material for mine site construction purposes, and specifically for reclamation cover systems [20,25,26]. For example, Rey et al. [27] tested in the laboratory and on field experimental cells, the use of desulfurized tailings as monolayer cover with an elevated water table system, and showed that the desulfurized tailings were efficient in limiting oxygen diffusion through the cover and preventing AMD generation. Qreshi et al. [28] confirmed using numerical modeling, the applicability of desulfurized tailings in an insulation cover with capillary barrier effects to prevent sulfide oxidation.

While the use of desulfurized tailings for mine reclamation received significant consideration, not as much research effort were invested on valorization of the desulfurization concentrate, i.e., the sulfide concentrate. With a much lower mass than desulfurized tailings, the sulfide concentrate remains a high-risk material due to its abundance of sulfide minerals. Investigated uses of desulfurization concentrate include its addition as a component of cemented paste backfill. Benzaazoua et al. [9] obtained high strength laboratory samples prepared with sulfide concentrate (12% S), mine tailings, cement and process water. However, cemented paste backfill prepared with high sulfide materials should be evaluated for potential loss of strength with time due to sulfate degradation [29]. The presence of valuable minerals and metals in the desulfurization concentrate was evaluated by Skandrani et al. [13], who attempted to enhance gold recovery to the concentrate when performing environmental desulfurization of an abandoned site's tailings. The work did not include the actual gold recovery from the concentrate.

Gold and other valuable elements may be concentrated along with the sulfides during environmental desulfurization. The concentrate produced by environmental desulfurization could become a promising source for metals or minerals that would otherwise be discarded with the tailings. If an economic value can be gained from the recovery of these metals, it could partly offset the costs for environmental desulfurization [8]. However, before this economical demonstration can be done, it is necessary to determine if it is possible to recover metals present in desulfurization concentrates. Particles that report to the desulfurization concentrate experienced the entire beneficiation circuit, where these particles were not recovered, and are carried with the remaining sulfides in the desulfurization concentrate. Their physical, chemical, superficial and mineralogical properties prevented previous recovery; therefore, it is important to investigate how much of these particles are suitable to a "last chance" recovery process.

The main objective of this project was to evaluate the potential gold recovery from desulfurization concentrates produced from tailings collected at three different gold mines in Abitibi-Témiscamingue, Quebec, Canada. The two major gold treatment processes

were investigated: cyanidation and gravity separation. Specific objectives include the gold deportment characterization of the desulfurization concentrate, and the effect of gold particle properties on the performance of cyanidation and gravity separation. A preliminary concentrate reprocessing flowsheet is proposed according to the results obtained for each mine site. This study is, to the authors' knowledge, one of the few that investigates specifically environmental desulfurization concentrates produced from acid generating tailings collected at the end of the beneficiation process.

## 2. Materials and Methods

### 2.1. Preparation and Characterization of Desulfurization Concentrates

2.1.1. Sampling and Preparation of Mine Tailings

Fresh tailings were collected as slurry in barrels at three different gold mine sites located in the Abitibi-Témiscamingue region in Quebec, Canada, and transported to RIME-UQAT laboratories. The three mine sites are active gold producers. Site A is an underground mine, and site B is an open pit mine, both processing ore by cyanidation. Site C is a polymetallic mine, with gravity separation, flotation, and cyanidation circuits, which generate gold ingots, copper and zinc concentrates. The objective of the desulfurization was essentially to produce sulfide concentrates for the remainder of the study, thus there was no attempt to optimize the sulfur content of the desulfurized tailings and flotation conditions from previous studies performed at RIME-UQAT were used [9,13]. The sulfur assay was different from one tailings to another, ranging from 1.3 to 13.4%, which influenced the mass of tailings to desulfurize to obtain at least 20 kg of concentrate, and the flotation equipment used. For sites A and C, 280 kg and 88 kg of tailings feed, respectively, were subjected to flotation to produce 52 kg and 27 kg of concentrate using the laboratory flotation column described in Section 2.1.2. For site B, 503 kg of tailings produced 19 kg of concentrate using the mobile pilot flotation plant described in Section 2.1.2

2.1.2. Production of Desulfurization Concentrate by Flotation

Flotation Column

The laboratory column flotation circuit used in this study is illustrated in Figure 1. It consists of: (i) a 190 L feed tank where the slurry is homogenized (Figure 1a), (ii) conditioning tanks of a capacity of 25 L each installed in series to adjust the activator $CuSO_4$, collector KAX (potassium amyl xanthate) and frother MIBC (methyl isobutyl carbinol) (Figure 1a) and (iii) a 4 m high by 7.6 cm diameter flotation column (Figure 1b). Reagents are injected in 25 L-conditioners using peristaltic pumps. The last conditioning tank feeds the flotation column. The feed pump flowrate is controlled through the froth level. Air is introduced near the bottom of the column by a bubble diffusor. Operating parameters such as pH and ORP of the slurry are measured in the feed and conditioning tanks. Concentrate is collected in the top launder and transferred to a concentrate barrel, while tailings are removed at the bottom of the column in tails containers.

The flotation parameters used for the desulfurization of tailings from mines A and C are presented in Table 1. The selection of these parameters was based on previous work performed by RIME-UQAT personnel on environmental desulfurization [9,13]. The operation of flotation was done over two days; the first day produced a rougher concentrate and tails, while the second day the rougher tails were reprocessed to produce scavenger concentrate and tails. The rougher and scavenger concentrates were combined into a single concentrate for gold recovery experiments.

Mobile Pilot Flotation Unit

The mobile pilot flotation unit is used when the number of tailings to desulfurize is over the flotation column's capacity (on a reasonable work time scale). The flotation unit (Figure 2) consists of: (i) a 200 L tank to prepare and homogenize the feed, (ii) two 50 L conditioners to add the reagents (i.e., collector, activator, frother), and (iii) a bank of 6 mechanical cells. By separating the bank in two, it was possible to perform rougher

(cells 1 to 3) and scavenger (cells 4 to 6) flotation concurrently. The activator CuSO$_4$ (300 g/t) was added in the first conditioner, while the collector KAX51 (180 g/t) and frother MIBC (50 g/t) were added in the second conditioner. Additional reagents were introduced after the third mechanical cell to enhance scavenging. Air flowrate was 5 L/min and agitation speed was 1500 rpm. Concentrate was collected by an automated paddle every 6 s.

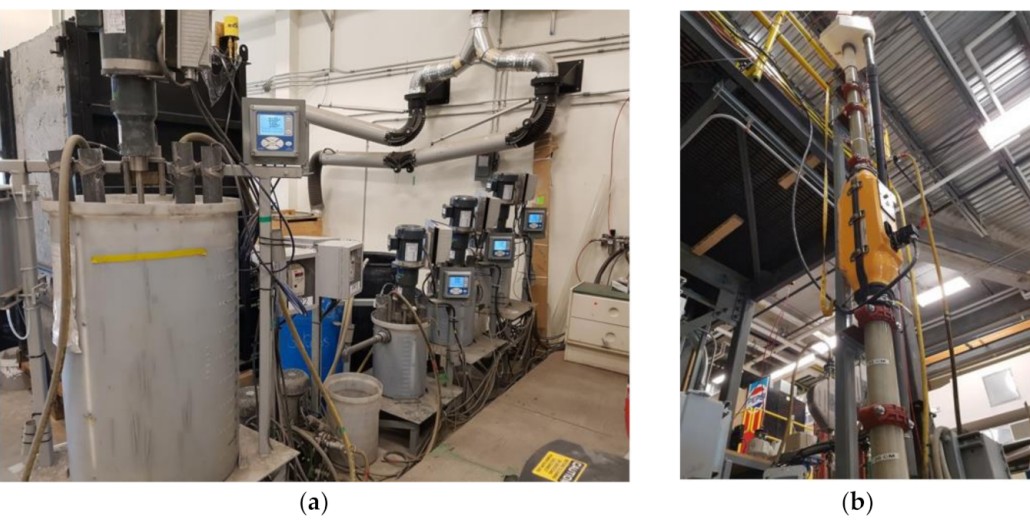

(**a**)  (**b**)

**Figure 1.** Photos of the RIME-UQAT column flotation circuit; (**a**) conditioning circuit, (**b**) flotation column.

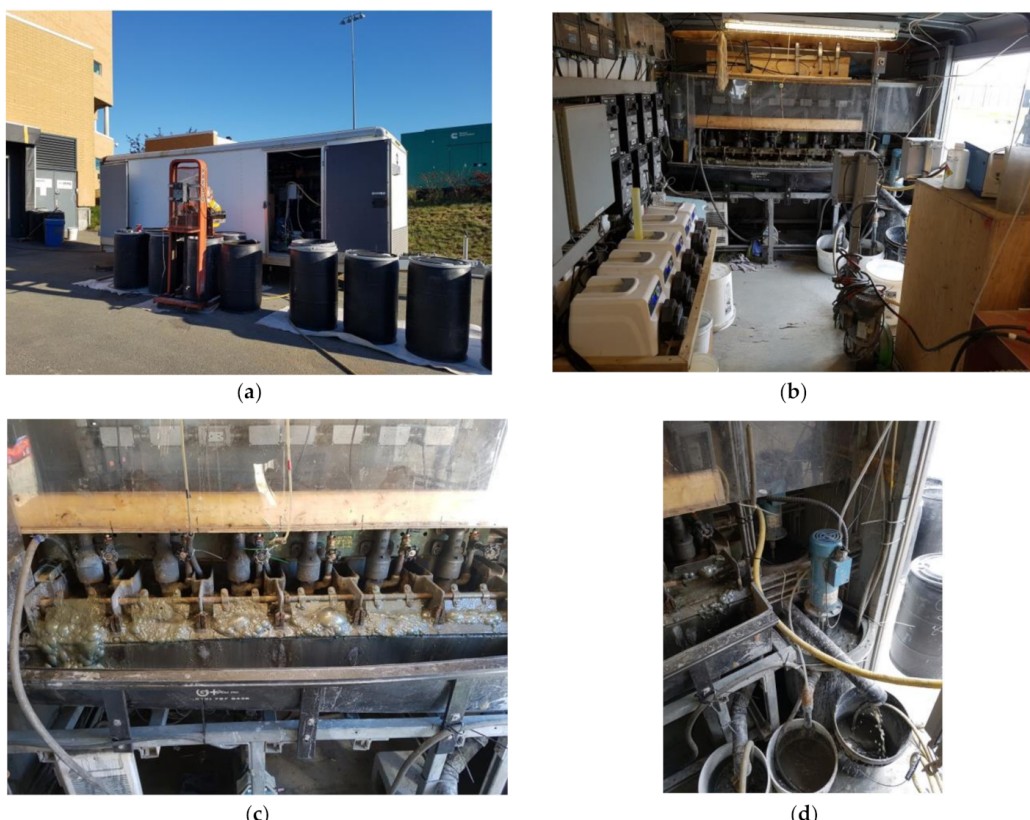

(**a**)  (**b**)

(**c**)  (**d**)

**Figure 2.** Mobile pilot flotation unit: (**a**) feed preparation area, (**b**) inside view with flotation bank at the rear and reagent pumps at the left, (**c**) flotation bank, (**d**) concentrate collection.

**Table 1.** Column flotation parameters used for environmental desulfurization of tailings from mines A and C.

| Parameter | Value |
| --- | --- |
| pH | Natural (slightly alkaline) |
| Feed slurry density | 15% (A), 25% (C) |
| Activator | 300 g/t $CuSO_4$, solution at 1% |
| Collector | 180 g/t KAX51, solution at 0.5% |
| Frother | 50 g/t MIBC, solution at 100% |
| Feed flowrate | 1.3 to 1.5 L/min |

2.1.3. Characterization of Desulfurization Concentrate

Once approximately 20 kg of sulfide concentrate from each tailings A, B, and C were produced, they were oven dried (40 °C), de-agglomerated, homogenized, and separated into 1 kg samples. A 1-kg bag was dedicated to physical, chemical and mineralogical characterization.

Relative density was evaluated with a helium pycnometer (Micromeritics Accupyc 1330—ASTM D5550-06). Particle size distribution was determined with a laser particle size analyzer (Malvern Mastersize S 2000). Chemical composition of the sulfide concentrates was obtained by ICP-AES (Perkin Elmer Optima 3100 RL) following complete acidic digestion ($HCl$, $HNO_3$, $HClO_4$ and $HF$) in microwave digestor. Total sulfur and total carbon were assayed with an induction furnace coupled to an infrared analyzer (ELTRA-CS-2000). The same procedure was used to monitor the sulfur distribution during desulfurization by flotation. Gold assays were acquired by pyroanalysis in an external laboratory.

X-ray diffraction (Bruker D8 Advance) was performed on micronized samples to estimate, semi-quantitatively, the main mineralogical composition, using Diffrac.EVA v 5.2.0.3 and TOPAS v 4.2 softwares for interpretation. Mineralogical observation of the samples was performed on polished sections (0.2 μm) coated with a carbon film using a sputter coater (Leica EM ACE600). The polished sections were sent to an external laboratory (IOS Services Géoscientifiques) for scanning electron microscope (SEM) analysis using the technology ARTGold™, which consists of an automated electron scan and rapid energy dispersion spectrometer (EDS) analyses on backscattered electron images. Gold-bearing particles are then re-examined at higher resolution to obtain a semi-quantitative chemical analysis.

*2.2. Gold Recovery by Cyanidation: Total and Kinetic*

Concentrates A, B, and C were first tested for total recovery by cyanidation. 500 g of concentrate were mixed with 600 mL of deionized water and agitated to form a 45% solids slurry in 2 L beakers (Figure 3). Lime ($Ca(OH)_2$) was added to the slurry to reach pH 12 (measured by pH sensor). Then, a 50 mL of a concentrated $CN^-$ solution was added to obtain a concentration of 1000 mg/L of $CN^-$ in the slurry. Agitation was maintained at 500 rpm for 58 h, the duration of the total recovery test. Frequent pH monitoring ensured that the pH stayed above 10.5 to prevent HCN formation and loss of $CN^-$. The three concentrates were tested in duplicate. After 58 h, the solids and liquid were separated by filtration, sampled and sent for gold analysis: by pyroanalysis in an external laboratory for solid samples, and by ICP-AES at UQAT for dissolved gold in liquid samples.

A similar protocol was performed for kinetic cyanidation tests. After slurry preparation as above, 10 mL of slurry samples were collected after 2, 4, 6, 8, 24, 48, and 58 h. The sampled volume was replaced by a cyanided replacement solution. Solids and liquids were separated by filtration and analyzed for gold concentration by pyroanalysis and ICP-AES, respectively. The concentration of $CN^-$ in solution and pH were monitored and adjusted, if necessary, after each sampling to maintain favorable cyanidation conditions ($CN^- > 150$ mg/L, pH > 10.5). Cyanide and lime concentration in the filtered solution were measured by titration. For cyanide measurements, typical silver nitrate titration with p-dimethylaminobenzalrhodanine 0.02% indicator was performed. For lime measurements,

an automatic titrator with alkalinity method (MA.303—Titr Auto 1.1 by Centre d'expertise en analyse environnementale du Québec) was used.

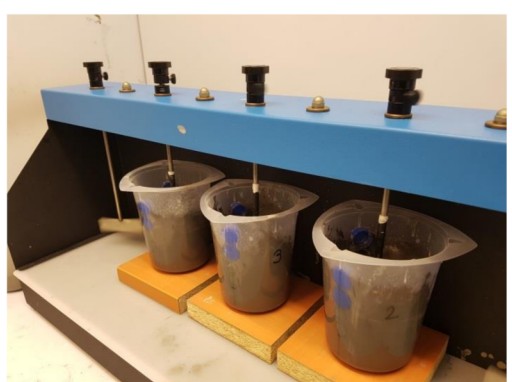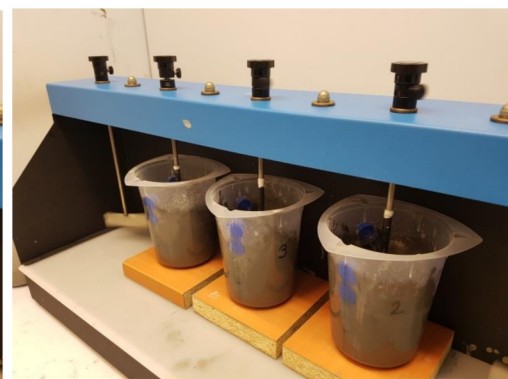

**Figure 3.** Laboratory set-up for total recovery and kinetic cyanidation tests.

### 2.3. Gravity Concentration

A two-step gravity concentration was performed to concentrate gold-bearing minerals: a first step with a Knelson concentrator, and then the Knelson concentrate was fed to a Mosley table for further concentration. One kg of desulfurization concentrate A, B, and C were mixed with water to form a 35% solids slurry that was introduced at a flowrate of 800 g/min into the laboratory scale 3-inch Knelson concentrator. Tap water was used as fluidizing water at 3 psi and 3.5 L/min. Centrifugal force was fixed at 60 G. Once all the feed was processed, several rinsing sequences were performed to recover the denser particles. The Knelson concentrate was then carried to the Mosley table, while the tails were filtered, dried, sampled and sent for gold analysis.

The Mosley table was fed with the Knelson concentrate. The operating conditions, i.e., table angle, shaking amplitude and frequency, fluidizing water flowrate, were set by the operator according to his experience to maximize gold separation. Because the 3 concentrates did not have the same specific gravity, these parameters were not identical for the 3 tests. Water flowrate was higher for A and B compared to C. The table concentrate was filtered, dried, weighed before sending for gold analysis. The Mosley table concentrate mass was too low for analysis, therefore it was diluted with pure silica to reach the required mass for analysis. This dilution was then considered when reporting the gold assays in the results section.

## 3. Results

### 3.1. Characterization of the Initial Tailings and Desulfurization Concentrate

3.1.1. Chemical Composition

Sulfur and gold assays for initial tailings (before environmental desulfurization by flotation), desulfurization concentrate (or sulfide concentrate), and desulfurized tailings are presented in Table 2. The most important parameters for this study are the sulfur and gold contents of the concentrates produced by environmental desulfurization. The sulfur content was above 20% for all concentrates, ranging from 23.0% for concentrate A to 34.5% for concentrate C. The gold content varied from 0.72 g/t for concentrate C, to 1.75 and 1.85 g/t for concentrates B and A, respectively. It can be noticed that the sulfur recovery was 89.8% for site A, 91.3% for site B, and 97.7% for site C. These recoveries are in the same range as previous studies on environmental desulfurization [12,18], therefore, the desulfurization concentrates produced were deemed suitable for the remainder of the experimental program.

**Table 2.** Sulfur and gold contents of initial tailings and environmental desulfurization concentrates.

| Tailings | Initial S Content in Tailings (%) | S Content in Desulfurized Tailings (%) | S Content in Desulfurization Concentrate (%) | Au Content in Desulfurized Tailings (g/t) | Au Content in Desulfurization Concentrate (g/t) |
|---|---|---|---|---|---|
| A | 6.00 | 0.80 | 23.0 | 0.56 | 1.85 |
| B | 1.32 | 0.12 | 25.2 | <0.08 | 1.75 |
| C | 13.4 | 0.50 | 34.5 | 0.09 | 0.72 |

Other elements of interest contained in the desulfurization concentrate, presented in Table 3, include Ni, which was present in low quantities in concentrates A (34 mg/kg) and B (127 mg/kg) and Cu, present in amounts between 795 and 2407 mg/kg, as well as elements representing neutralizing minerals (Ca, Mg).

**Table 3.** Selected chemical composition of desulfurization concentrates.

| Concentrate | Fe (%) | Ni (mg/kg) | Cu (mg/kg) | Ca (mg/kg) | Mg (mg/kg) |
|---|---|---|---|---|---|
| A | 20 | 34 | 2407 | 28,684 | 7985 |
| B | 21 | 127 | 1282 | 13,140 | 12,010 |
| C | 28 | <5 | 795 | 4027 | 1710 |

3.1.2. Physical Characterization

Relative density was 3.15 for concentrate A, 3.19 for concentrate B, and 3.58 for concentrate C. The feed to the desulfurization process had a density below 3 for all materials. The higher density in the three concentrates compared to the mine tailings (feed) confirms the concentration of dense minerals, i.e., sulfides, and corroborates with the measured concentrations of S and Fe in the concentrate (Tables 2 and 3).

Based on the particle size distribution, presented in Figure 4, concentrate B was slightly finer than the other two concentrates, while concentrate C extended more towards the coarser fractions in the upper portion of its distribution. The diameter of 80% passing, or D80, was 30.1 μm, 22.1 μm and 51.5 μm for concentrates A, B, C, respectively. D50 was 13.5 μm for concentrate A, 8.2 μm for concentrate B, and 17.9 μm for concentrate C. These fine particle sizes are explained by the previous beneficiation processes that required liberation, and by the desulfurization process by flotation, which is more efficient in particle sizes in the range of 10 to 100 μm [30]. These fine particle sizes are expected to be detrimental to the gravity concentration processes.

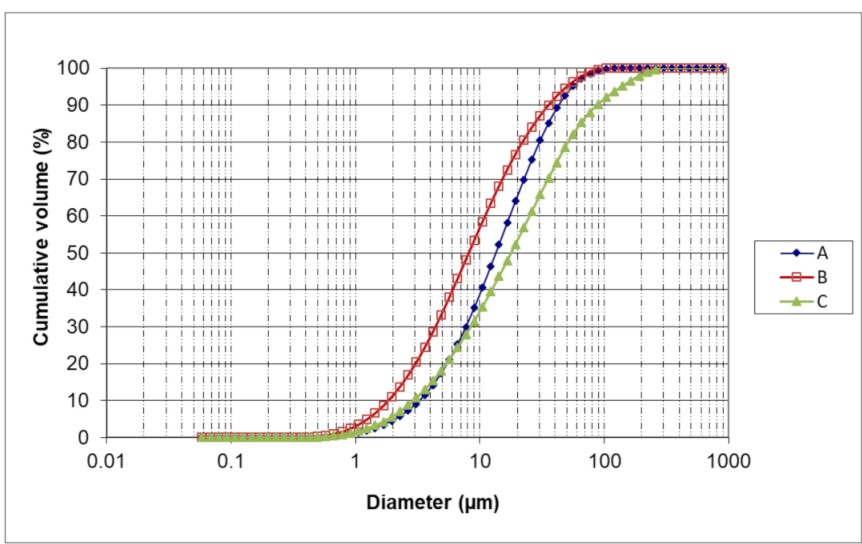

**Figure 4.** Particle size distributions of concentrates A, B, and C.

### 3.1.3. Mineralogical Characterization

Pyrite content in the concentrates was evaluated by XRD. Concentrates A and B had approximately 30 and 33% pyrite, respectively. Other major minerals include quartz and diopside for concentrate A, and albite and quartz for concentrate B. Concentrate C had 57% pyrite and 32% quartz. XRD patterns and semi-quantitative interpretation are presented in Figure 5.

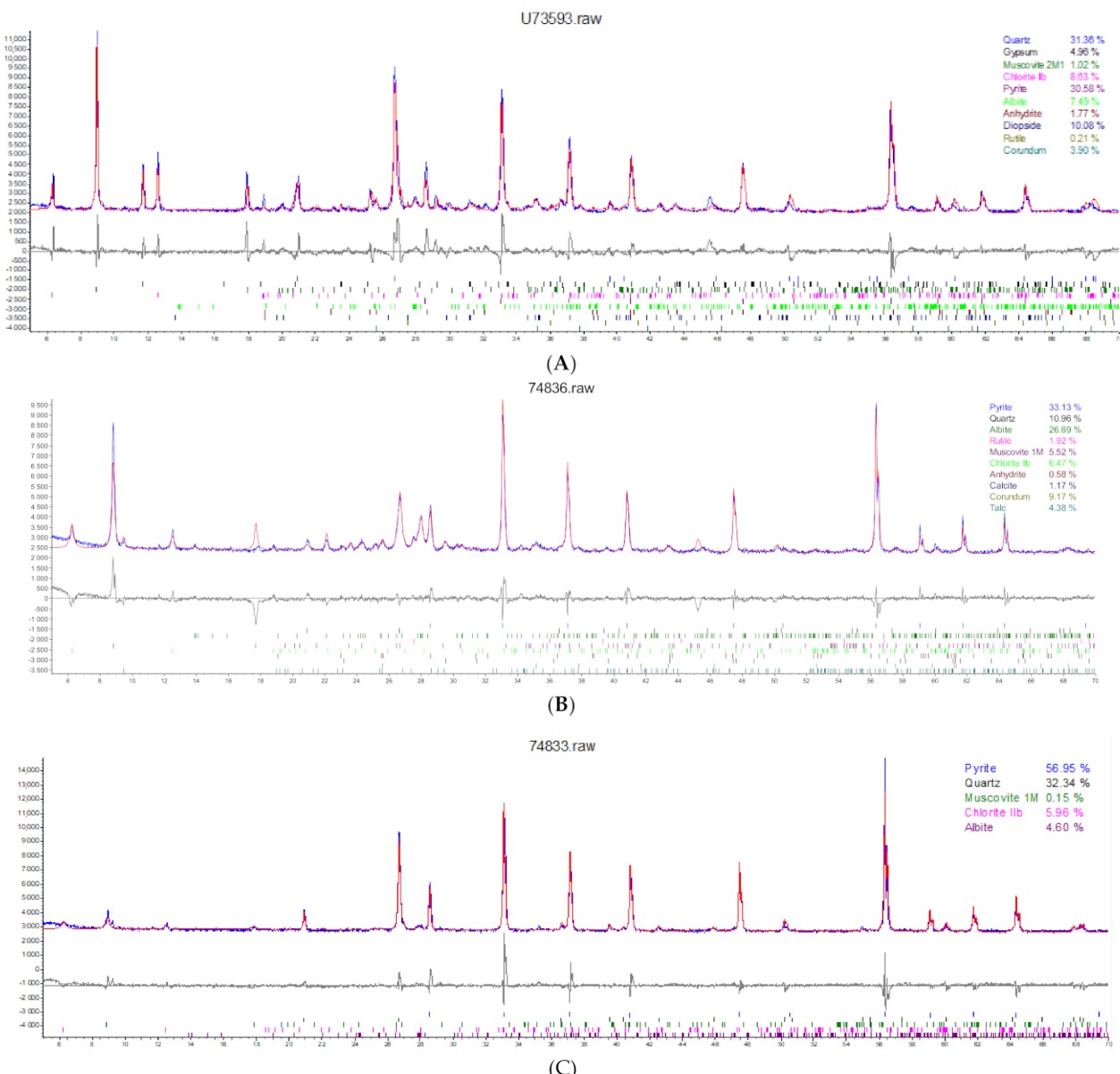

**Figure 5.** XRD spectra and interpretation for concentrates (**A–C**).

Prior to gold recovery tests, the gold particles liberation and mineral associations were investigated to provide hints on the potential success of recovery methods. The mineralogical assessment was considered statistically representative only for concentrate B sample due to the limited number of gold particles found in polished sections for concentrates A and C. However, information on concentrates A and C are also presented with a cautionary notice that they may not represent the entire lot.

Gold particles were identified as locked at 99% in concentrate B. Observed particles in concentrates A and C indicated that Au particles seemed to be also locked. The hypothesis that gold is associated with sulfides is also confirmed with the mineralogical analysis. The

main associated mineral was pyrite, with 94% of gold encapsulated within pyrite particles in concentrate B. For concentrates A and C, pyrite was also identified as the main gold particle host mineral. This observation is concordant with the desulfurization process which aimed at recovering pyrite and other sulfides as a bulk flotation concentrate. Measurements of gold particle size revealed that most gold particles are very fine, below 3 μm.

Gold mineral composition was evaluated for concentrate B, as shown in Figure 6. Petzite ($Ag_3AuTe_2$) is the main gold phase observed (73%). Non-stochiometric calaverite (Au-Te mineral) was also observed in lower amounts (11%), while electrum (Au-Ag) accounts for 12% of the gold particles.

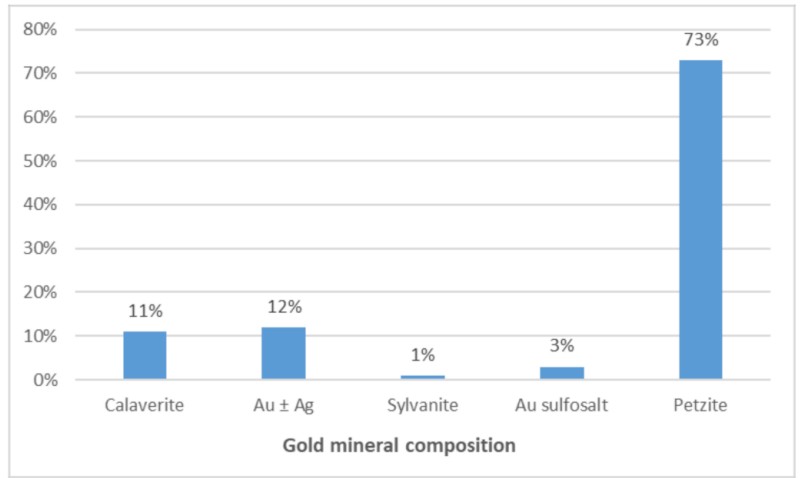

**Figure 6.** Gold deportment in concentrate B.

The fact that gold particles are very fine (<3 μm) and mainly encapsulated signifies that loss in the concentrator tailings was due to a lack of liberation and their low particle size, and not to inefficient treatment processes at the mine sites. The recovery of such fine and locked particles is expected to be challenging, for both chemical (cyanidation) and physical (gravity separation) approaches.

### 3.2. Cyanidation

Gold recoveries (total and kinetic) calculated from the cyanide solutions obtained for concentrates A, B and C are presented in Figure 7. For the three concentrates, both duplicates behaved similarly, and in the typical pseudo-first order reaction of gold dissolution by a cyanide solution [31]. The first test (total recovery, identified by the green bar) was performed to obtain the total recovery after 58 h of cyanidation. Results were different from one concentrate to the other. Concentrates A and C had moderate Au recoveries of 46% and 59%, respectively, while Au recovery from site B was very low with only 10.5% of Au recovered after 58 h. The initial cyanide concentration of 1000 mg/L was expected to decrease over the course of the test, down to approximately 150 mg/L at the end of 58 h. Below this concentration, the reaction rates can drastically be reduced. The final $CN^-$ concentrations were between 115 and 135 mg/L for concentrates A and B, while a final concentration of 290 mg/L was recorded for concentrate C, which indicated that the low to moderate Au recoveries obtained were not due to an overconsumption of cyanide ions by some competitive species and a lack of cyanide ions at the end of the reaction.

The kinetic cyanidation test evaluated the evolution of the reaction rate throughout the test. Concentrate A showed a moderate increase in recovery in the first 10 h, then leveled off after 24 h at recoveries between 50 and 55%. The gold concentration in the solution for the kinetic test at 58 h was 0.52 mg/L (and 0.54 mg/L for the duplicate). Concentrate B showed again a very low affinity for gold recovery by cyanidation. The recovery and gold concentration in the solution remained relatively constant after 8 h at values of 13 to 16% and 0.14 mg/L, respectively. Concentrate C was able to improve its final recovery

at 58 h up to 74 and 78%, with a Au concentration of 0.27–0.28 mg/L in solution. Again, most of the gold dissolution occurred in the first 24 h of the test. Kinetic cyanidation was expected to yield higher final recoveries compared to the total recovery test because the $CN^-$ concentration was adjusted to remain above 150 mg/L at every sampling time step. Concentrate A required $CN^-$ adjustments with a higher volume of $CN^-$ solution (51 and 61 mL for A and A-dup) compared to concentrate B (41 and 33 mL for B and B-dup, respectively) and concentrate C (18 and 27 mL for C and C-dup, respectively).

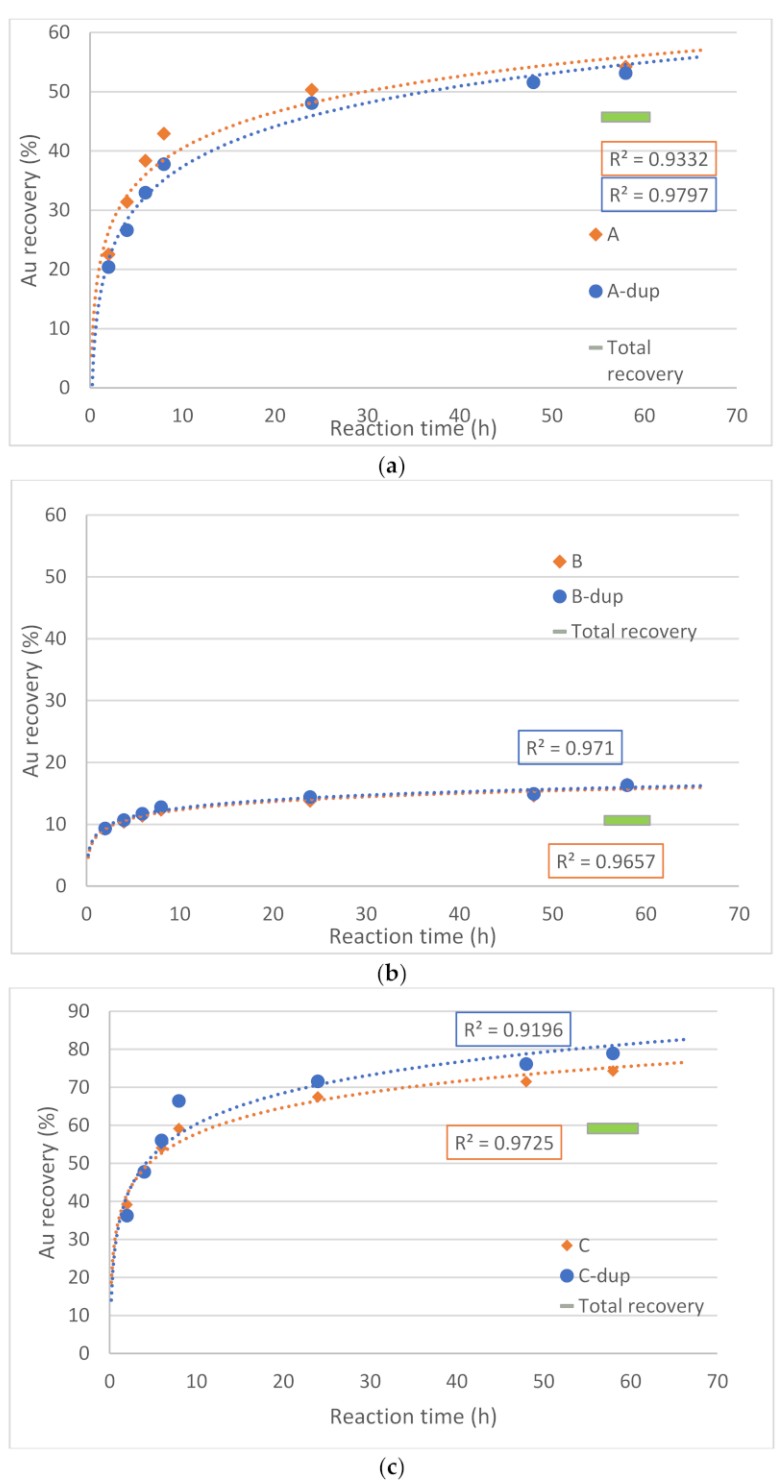

**Figure 7.** Results from kinetic and total cyanidation tests for (**a**) concentrate A, (**b**) concentrate B, (**c**) concentrate C, including the duplicate kinetic cyanidation test.

The presence of copper can influence the performance of cyanidation by reducing the availability of $CN^-$ to complex gold, a well-known phenomenon called preg-robbing [31]. It was observed that $CN^-$ consumption was higher for concentrate A, which contained the highest concentration of Cu (2407 mg/kg). Concentrate B contained 1282 mg/kg of Cu which could have also complexed a portion of $CN^-$ and reduced the available $CN^-$ concentration. Concentrate C, which had the lowest Cu content (785 mg/kg), exhibited the lowest $CN^-$ consumption, and the highest gold recovery (59%). However, the Cu contents in the tested concentrates are lower than 0.5%, which was identified as a criterion for cyanidation of gold ores with minimal cyanide consumption and related economic and technical issues [32].

The locked gold particles can explain the very low recoveries encountered with concentrate B. Due to limited liberation, the cyanide solution had low contact with the gold particles and was unable to dissolve them. The mineralogical investigation being not statistically representative for concentrates A and C, it is not possible to extrapolate the same amount of encapsulation to gold particles contained in concentrates A and C. Their better performance in cyanidation may be related to a slightly better liberation that was not possible to observe due to the low number of gold particles in the samples sent to the mineralogical analysis.

Pyrite may also have an influence on cyanidation efficiency. Indeed, soluble sulfides can reduce the gold recovery rate by passivating the gold particle surfaces under atmospheric conditions [32–34], but under oxygen-enriched conditions, gold leaching rates can be accelerated [35]. Strong aeration with oxygen could be tested eventually to improve the gold recovery considering the significant pyrite content of the sulfide concentrates.

### 3.3. Gravity Recovery

Gravity recovery was performed on the desulfurization concentrates in a two-step process. Indeed, the mineralogical assessment suggested that gold particles were very fine and encapsulated, which may not be the ideal conditions for gravity recovery. Nevertheless, the process was tested due to its ease of operation and the lack of chemical reagents. The concentrates were first passed through a Knelson concentrator, then the Knelson concentrate was further treated on a Mosley table.

Gold content in the initial samples is compared to the concentrates obtained by the centrifugal concentrator and the separation table in Table 4. The duplicate samples did not have similar results for all products; it is expected that nugget effects are more prominent in gravity concentration processes. The nugget effect may also be related to the feed material, where 1 kg of desulfurization concentrate was sampled from the total concentrate produced. With such low gold content, heterogeneities may occur even if the homogenization and sampling process was performed correctly. An issue occurred with concentrate A manipulations where approximately 6% (or 60 g) of the total mass of the sample was lost. The problem was resolved before experiments for the other concentrates and nonsignificant mass loss was recorded for concentrates B and C.

**Table 4.** Gold assays in initial concentrate and Knelson and Mosley table concentrates.

| Concentrate | Initial Au (g/t) | Knelson Concentrate Au (g/t) | Mosley Table Concentrate Au (g/t) |
|:---:|:---:|:---:|:---:|
| A | 1.85 | 4.27 | 7.91 |
| A—dup | 1.85 | 4.18 | 12.24 |
| B | 1.75 | 3.59 | 7.22 |
| B—dup | 1.75 | 2.59 | 5.16 |
| C | 0.72 | 2.25 | 23.637 |
| C—dup | 0.72 | 1.41 | 5.92 |

The gravity process produced a final gold concentrate of 7.9 g/t (concentrate A) and 12.2 g/t (concentrate A-duplicate) from desulfurization concentrate A. Calculated recovery was 67% (53% for duplicate) at the Knelson step, and 54% (49% for duplicate) at the Mosley table step. The overall gold gravity recovery for concentrate A was 36% and 26% for the duplicate.

For concentrate B, the final gold product assayed 7.2 and 5.2 g/t. Knelson recovery was evaluated at 27% (19% for duplicate) while Mosley table recovery was calculated as 24% and 27% (for duplicate). Global gravity recovery was low at 6% (5% for duplicate). Concentrate B was the one with the lowest mass at the end of gravity separation, with 7.74 g and 12.55 g for the duplicate experiment.

Concentrate C had the lowest initial Au content and reached a Knelson concentrate with 2.3 and 1.4 g/t Au grade. Both samples had masses below 20 g before dilution with silica. Knelson recovery was 57% and 29% for the duplicate sample. Concentrate C showed the most significant gold assay difference between the duplicate samples, with a final gold concentrate of 23.6 g/t and 5.9 g/t. Table recovery was affected by the high grade measured in the first sample; a negative recovery was calculated, while a 50% recovery was obtained for the duplicate sample with a 5.92 g/t grade. This result hints that the 23.6 g/t grade is probably overestimated, and the 5.92 g/t assay seems more realistic. The overall gravity recovery for the duplicate sample was 14.6%.

It appears that concentrate B was the least receptive to gravity recovery, again in concordance with the mineralogical assessment results. Concentrate A performed better, and concentrate C showed modest grades and recoveries, although difficult to state clearly due to assay overestimation. All these gold concentrations are lower than those encountered in an industrial setting. It is also important to keep in mind that the feed consisted of mill tailings that were concentrated by desulfurization prior to gravity recovery, therefore the gold particles were once rejected by an efficient industrial gold recovery process and quite challenging to concentrate in the present study.

## 4. Discussion

Environmental desulfurization has proven its applicability to reduce the acid generation potential of tailings [1]. To fully consider the integrated tailings management concept, it is essential to provide valorization options for both desulfurized tailings and desulfurization concentrate. Use of desulfurized tailings as part of tailings storage facility closure and reclamation was investigated by several researchers [20,22,23,26,27,36]. Recovery of valuable elements from desulfurization concentrate was not studied as much.

Characterization results for the three concentrates produced by desulfurization suggested that the potential for gold recovery was marginal, due to (i) the low Au content (0.7 to 2 g/t), (ii) gold particles being mainly encapsulated in pyrite, (iii) the fine particle size of the gold. Nevertheless, it was attempted to recover a portion of the gold as a way to valorize an otherwise final tailings stream. It would be unreasonable to compare the gold grades and recoveries obtained in this study with traditional beneficiation circuits, since the feed material is significantly different.

Concentrate B was the least successful in terms of gold recovery, both for cyanidation and gravity recovery. The chemical analysis did not reveal a significant number of other elements that could be valuable; however the list of elements was not exhaustive. More detailed analyses could be performed to assess the presence of critical and strategic metals, such as rare earth elements, that were not analyzed in this study. Furthermore, other valorization routes could be investigated for this concentrate. Site B is expected to expand to an underground mine where cemented paste backfill, made in part with desulfurization concentrate, could be used to fill the excavated stopes [37,38].

Approximately 50% of the gold from concentrate A was recovery by cyanidation, and 26 to 36% by gravity separation. While these amounts would probably not deserve a distinct concentration circuit, it would be interesting to investigate the possibility of re-introducing the desulfurization concentrate into the existing cyanidation circuit, as shown in Figure 8.

This loop becomes an additional opportunity to recover gold that would otherwise exit the mineral processing plant. A second opportunity that could be valuable for site A if reprocessing through the cyanidation circuit is not possible would be to add a small gravity concentration circuit for the desulfurization concentrate, where the gravity tails can feed the cemented paste backfill plant along with desulfurized (or not) tailings.

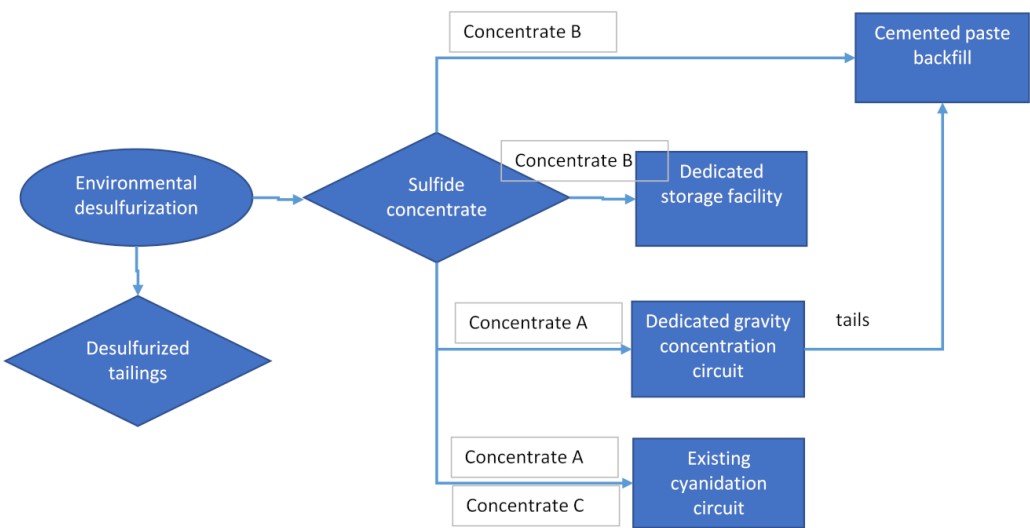

**Figure 8.** Proposed flowsheet highlighting the valorisation options for sulfide concentrate produced via environmental desulfurization.

Concentrate C had the lowest gold concentration, but showed significant recoveries by cyanidation (above 70%). A circulating loop between the desulfurization concentrate and the existing cyanidation circuit would be a suggested approach to recover additional gold, as proposed for site A.

Sites A and C have cemented paste backfill plants that could also integrate the desulfurization concentrates, and/or the tails from a gravity concentration circuit.

This project highlighted the importance of doing the laboratory experiments even on low gold grade desulfurization concentrates. Indeed, concentrate C, with only 0.7 g/t Au, showed the highest recovery by cyanidation of the three tested concentrates. The mineralogical assessment, and most specifically gold deportment, is also a necessary tool to anticipate and interpret the results. Unfortunately, the statistical requirements were not met to get a representative analysis for concentrates A and C, however clues on the gold behavior were provided.

## 5. Conclusions

The potential to recover gold from desulfurization concentrates originating from three different mine sites was evaluated. To do so, three gold mine sulfide concentrates obtained by environmental desulfurization were characterized and two typical gold recovery processes (cyanidation and gravity separation) were tested.

Physical, chemical and mineralogical characterization identified that the concentrates had a gold content between 0.7 and 1.9 g/t, sulfur contents between 23 and 34.5%, and a fine particle size distribution (D80 < 52 µm). The assessment of Au-bearing particles was statistically representative for concentrate B only, due to the limited number of Au particles found in the polished sections for concentrates A and C. Gold particles were mostly (94%) associated with, and locked in (99%), pyrite; indicating the potential refractoriness of Au-bearing minerals and the associated challenges related to its recovery, especially for concentrate B. Cyanidation showed gold recoveries of 50 to 55% for concentrate A, 13 to 16% for concentrate B, and 74 to 78% for concentrate C. Gold-bearing particle liberation and copper concentration in the desulfurization concentrate were identified as parameters negatively affecting Au recovery by cyanidation. Gravity separation was less effective

to recover gold, mostly below 50%, due to the fine particle size of the desulfurization concentrates. A proposed flowsheet for sites A and C includes the recirculation of the desulfurization concentrate into the existing cyanidation circuit to minimize the gold losses to the final tailings.

Finally, this work highlighted the interest in characterization of the desulfurization concentrate to identify potential valuable elements. Although the focus was on gold, a similar approach could be used for investigations of other metals or minerals of value, such as rare earth elements and critical minerals. This work is also a demonstration of the overlap of the fields of mineral processing and environment. Indeed, mineral processing techniques are used to help solve an environmental issue, AMD, while this environmental solution can provide further production benefits for the mining operation.

**Author Contributions:** Conceptualization, O.A., M.L., I.D. and L.C.; methodology, O.A., M.L., I.D. and L.C.; formal analysis, O.A. and M.L.; investigation, O.A. and M.L.; writing—original draft preparation, I.D., O.A. and M.L.; writing—review and editing, O.A., M.L., I.D. and L.C.; supervision, I.D. and L.C.; project administration, I.D.; funding acquisition, I.D. All authors have read and agreed to the published version of the manuscript.

**Funding:** This research was funded by a NSERC Collaborative Research Development Grant awarded to I. Demers, grant number CRDPJ 519789-2017, and industrial partners.

**Data Availability Statement:** Not applicable.

**Acknowledgments:** The authors want to thank the industrial partners involved in the CRD Grant and in RIME-Polytechnique-UQAT. The personnel of UQAT laboratories are thanked for their help in the experiments.

**Conflicts of Interest:** The authors declare no conflict of interest.

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
