# Peer review of "Gold Recovery from Sulfide Concentrates Produced by Environmental Desulfurization of Mine Tailings"

_minerals, doi:10.3390/min12081011_

Round 1
Reviewer 1 Report
Authors state at the beginning, that the tailings generate acids, so they are harmfull for the environment. but in Table 1 the pH is natural (slightly alkaline) why is that?
The flotation parameters are xetreamly different, one concentrate was collected 30% and the other one 3,7%, why is that?
(169) laser particle size analyzer is in most cases working with wet samples, and the samples were dried and then applied to the analyser. it can cause particle size distribution error due to creation of bigger particles duruing drying.
(343) if Autorhs know that pyrite can influence the cyanidation efficiency, have they applied the extensive oxigenation of the suspensions during the leaching? It would be good to have a leaching repeated with oxigenation so it could be determined was low Au leaching effect of lack o oxygen and pyrite effect, or not.
also buringin of the samlples would be interested to libarete the Au for comparation.
Reviewer 2 Report
Journal: Minerals (ISSN 2075-163X)
Manuscript ID: minerals-1837688
Type: Article
Title: Gold recovery from sulfide concentrates produced by environmental desulfurization of mine tailings
Authors: Olivier Allard, Mathieu Lopez, Isabelle Demers, Lucie Coudert
Section: Mineral Processing and Extractive Metallurgy
Special Issue: Towards a Sustainable Management of Mine Wastes: Reprocessing, Reuse, Revalorization and Repository, Volume II.
Comments for authors:
The work proposed by the authors addresses a very topical and important issue, not only economically but also environmentally. The treatment of mining waste and mining liabilities is a very serious problem today.
The Abstract is well written and clearly denotes the objective of the work; moreover, the authors' mastery of the field is evident.
The introduction is well argued and an extensive bibliographical citation is provided.
The results, Discussion and conclusions are well presented and argued.
However, a few comments are desired that will help to refine the structure and substance of the paper. These recommendations appear below:
Abstract: it is recommended to mention which method was employed in the mineralogical characterisation of the studied samples. Fix.
Abstract: It is recommended to add a paragraph at the end of the Abstract commenting in which field of mining and/or metallurgy the results and experiences obtained through this research could be applied. Fix.
Figures 1, 2 and 3. It is evident that the photographs presented belong to one of the authors, in that case, it is not necessary to write the name of the author (M. Lopez), as in that case it should also appear in the "References" section. Please delete.
Table 3. In columns 4, 5 and 6, replace the comma with a full stop in the numbers. Fix.
Subsection 3.1.3 'Mineralogical characterisation'. Why have the authors not added X-ray diffraction patterns?
Reviewer 3 Report
Dear Editor-in-Chief
I read the manuscript thoroughly. An environmental-friendly method in order to evaluation of gold recovery from exist tailing was developed. I think it will require some works to reorganize the existing manuscript. Hereby some comments are provided to enhance the manuscript.
- - In part 2.2, the methods used for measurement of cyanide and lime concentrations is necessary to be mentioned.
- - The reason/s for differences of water flowrates (part 2.3, line 221) is/are recommended to be explained.
- - The method used for specifying gold mineral composition (part 3.1.3, line 284) is recommended to be mentioned.
- - The phenomenon described in page 11, preg-robbing, can be disclosed by a simple FTIR analysis. The results will help to accurately describe the issues facing and to develop the result demonstrations.
- - Based on obtained results, the priorities should be indicated in Fig. 8.
